# Spice Up Your Kidney: A Review on the Effects of Capsaicin in Renal Physiology and Disease

**DOI:** 10.3390/ijms25020791

**Published:** 2024-01-08

**Authors:** Michela Musolino, Mario D’Agostino, Mariateresa Zicarelli, Michele Andreucci, Giuseppe Coppolino, Davide Bolignano

**Affiliations:** 1Nephrology and Dialysis Unit, Magna Graecia University Hospital, 88100 Catanzaro, Italy; mikymusolino@gmail.com (M.M.); mariodagostino1092@gmail.com (M.D.); andreucci@unicz.it (M.A.); gcoppolino@unicz.it (G.C.); 2Department of Health Sciences, Magna Graecia University, 88100 Catanzaro, Italy; mteresa.zicarelli@gmail.com; 3Department of Medical and Surgical Sciences, Magna Graecia University, 88100 Catanzaro, Italy

**Keywords:** capsaicin, kidney, hypertension, kidney disease, renal cancer

## Abstract

Capsaicin, the organic compound which attributes the spicy flavor and taste of red peppers and chili peppers, has been extensively studied for centuries as a potential natural remedy for the treatment of several illnesses. Indeed, this compound exerts well-known systemic pleiotropic effects and may thus bring important benefits against various pathological conditions like neuropathic pain, rhinitis, itching, or chronic inflammation. Yet, little is known about the possible biological activity of capsaicin at the kidney level, as this aspect has only been addressed by sparse experimental investigations. In this paper, we aimed to review the available evidence focusing specifically on the effects of capsaicin on renal physiology, as well as its potential benefits for the treatment of various kidney disorders. Capsaicin may indeed modulate various aspects of renal function and renal nervous activity. On the other hand, the observed experimental benefits in preventing acute kidney injury, slowing down the progression of diabetic and chronic kidney disease, ameliorating hypertension, and even delaying renal cancer growth may set the stage for future human trials of capsaicin administration as an adjuvant or preventive therapy for different, difficult-to-treat renal diseases.

## 1. Introduction

Capsaicin is a chemical molecule abounding in the seeds and berries of red peppers and chili peppers (genus *Capsicum*), being responsible for the hot, irritant flavor and taste of these spices [1]. Red peppers and chili peppers have been used over the history as unique flavoring agents as aphrodisiacs but also as true medicines or natural remedies. Capsaicin was first discovered in 1816 by P.A. Bucholtz, but first extracted in 1876 by J.C. Thresh, who finally gave it its renown name. Several beneficial properties have been attributed to capsaicin based on its nociceptive, anti-inflammatory, immune, and anticancer effects, thereby suggesting its possible therapeutic application in different disease settings [2]. Currently, capsaicin is widely employed as a topical, analgesic drug for treating neuropathic as well as inflammatory pains, such as in diabetic neuropathy, post-herpetic neuralgia, rheumatoid arthritis, and osteoarthritis [2]; non-pain indications for topical capsaicin include the treatment of itch, psoriasis, and allergic rhinitis [3].

Interestingly, capsaicin also plays various roles in the kidneys, spanning from the regulation of renal physiology to potential benefits, particularly regarding kidney diseases. In this review, we aimed to summarize the key evidence regarding the putative role of capsaicin on kidney function and pathophysiology and its possible therapeutic applications for treating some renal diseases.

## 2. Capsaicin—Biochemical Properties and Mechanism of Action

Capsaicin is an organic compound with a phenolic structure (8-methyl-N-vanillyl-6-nonenamide). It is insoluble in water, containing a vanillyl group (the head), an amide group (the neck), and a fatty acid chain (the tail) (Figure 1).

Up to 94% of the administered capsaicin is well-absorbed either by an oral or topical route [3]. Capsaicin acts as the agonist of the transient voltage-gated receptor potential vanilloid-1 (TRPV1), also known as the capsaicin receptor, which was firstly discovered in the rat dorsal root ganglia [4]. TRPV1 is a tetrameric channel, with N and C termini of each subunit located intracellularly [4]. The affinity between capsaicin and the TRPV1 channel is highly selective and potent, being well-studied on a three-dimensional level by Fan Yang and Al. [5], who firstly observed that the bond between the molecule and the receptor takes a ‘tail-up, head-down’ configuration, through interactions between the vanillyl group and the S4-S5 linker. TRPV1 is a non-selective cation channel; when activated, sodium and calcium ions flow into the cell to depolarize nociceptive neurons, promoting an action potential which underlies the sensation of spiciness [4].

This receptor is located on the plasma membrane, mainly expressed at the endoplasmic reticulum level, playing a key role in intracellular calcium homeostasis [6]. TRPV1 releases calcium from intracellular stores and controls the calcium amount inside the mitochondria [7].

TRPV1 is a polymodal receptor and can be activated by many stimuli beside capsaicin, such as high temperature or acidosis, as well as by some toxins or compounds found, for example, in wasabi or mustard [3]. After chronic exposure to capsaicin, the activity of TRPV1 decreases (i.e., the desensitization phenomenon), due to an elevation of intracellular calcium levels. This phenomenon both has a protective mechanism to avoid calcium overload toxicity and contributes to the analgesic effects attributed to capsaicin [8]. On these premises, TRPV1 is currently considered as a target for nociception, able to increase the influx of calcium ions in dorsal root ganglion neurons [9,10]. Accordingly, mice lacking the TRPV1 receptor exhibit an impaired thermo-mechanical acute pain sensation [11].

Besides its role on TRPV1 activation, capsaicin may regulate other ions’ flux, reactive oxygen species production, and cellular membrane fluidity [12]. For these reasons, this compound has also extensively been studied as a powerful antioxidant and anti-inflammatory agent.

TRP vanilloid channels abound in different human tissues, such as the skin [13], the endothelium [14], and even the kidney (renal tubule) [15]. This latter observation has prompted the interest of investigating the possible renal effects played by capsaicin under either physiological or pathological conditions.

## 3. Functional and Structural Effects of Capsaicin on the Kidney

As briefly mentioned before, the TRPV1 channel is largely expressed in the tubules of the renal cortex and medulla [15]. However, besides the TRPV1, capsaicin may target other receptors which modulate various renal activities (Figure 2).

As an example, the transient receptor potential cation channel member 6 (TRPC6) is predominantly expressed in podocytes, and its functional alterations have been called into question in the pathogenesis of various proteinuric kidney diseases [16]. TRPV4 receptors (TRP vanilloid 4 receptor channels) promote the influx of calcium into tubular and endothelial cells [16]. Both the TRPV4 and TRPV1 receptors are crucial in maintaining the functionality of both the endothelial barrier and the regulation of cell junctions [17,18] and are likely involved in the functionality of the glomerular filtration barrier, as well [19]. Notoriously, podocyte injury or TRPV1-V4 derangement leads to different glomerular diseases with consequent massive proteinuria [20]; hence, therapeutic modulation of these receptors by capsaicin could theoretically represent a potential therapeutic target for this kind of renal disease. Likewise, the TRP melastatin channels (TRPM2) are present in the cytoplasm and intracellular organelles of renal tubular cells, and their inhibition may improve the outcome of experimentally induced renal ischemia [21]. Capsaicin was also proved to modulate the integrity of tight junctions of Canine Kidney-C7 epithelial cells [22], enhancing their permeability to poorly adsorbable molecules. By the same token, derivates of capsaicin with lower lipophilicity can increase the permeability of hydrophilic compounds by opening the tight junctions for a shorter time than capsaicin, proving that lipophilicity could interfere within tight junction activity. Notably, this mechanism seems to be independent from TRPV1 activity and calcium ions’ flux [22]. TRP receptors are also involved in the vaso-regulation of kidney vessels [23]. Accordingly, capsaicin promotes renal vessel vasodilation in a concentration-dependent manner by binding these receptors [23]. TRPV4 promotes vascular relaxation in large renal arteries, in renal conduit arteries, and in medullary vasa recta, while TRPV1 displays a narrower distribution and more undefined vasoactive effects [15]. Specifically, TRPV1 regulates pre-glomerular vascular resistance, thereby explaining why acute capsaicin administration may temporarily increase the estimated glomerular filtration rate (eGFR). The activation of TRP vanilloid channels produces a calcium influx into endothelial cells, triggering the relaxation of different renal arterioles depending on the receptor localization. In fact, while TRPV1 vasodilates large renal resistance arteries with no effects on renal conduit arteries or renal vasa recta, TRPV4 induces the relaxation of renal conduit arteries and smaller intrarenal resistance arteries [23].

On top of these hemodynamic effects, capsaicin may also regulate diuresis and natriuresis [24,25]. Such an effect would rely upon a decrease in the perfusion pressure, driven by an enhanced vaso-relaxation mediated by the calcitonin gene-related peptide (CGRP) and the substance P, two largely acknowledged powerful vasodilators [24]. The key role of CGRP was also confirmed by another experimental model [21]. A wire myograph was employed to measure isometric tension changes in the renal tubules of mice while mechanic stimuli were applied to the pelvicalyceal junction. Capsaicin administration to those mice promoted slowed and enlarged spontaneous phasic contractions. The negative chronotropic effects on those contractions were probably due to the release of CGRP as induced by TRPV1 activation by sensory nerves. Interestingly, the unilateral activation of TRPV1 sensory nerves innervating the renal pelvis of rats stimulated bilateral diuresis and natriuresis independently from the CGRP concentration [26]. Finally, in another experiment, the infusion of either capsaicin or CGRP prevented the increase in perfusion pressure as induced by norepinephrine [27], while the preventive administration of TRPV1 antagonists could blunt the hemodynamic effects induced by capsaicin [24], causing a greater vasodilation in afferent arterioles than the efferent ones. This latter observation suggests that under physiological conditions, TRPV1 may play a fundamental role in the protective, auto-regulatory mechanisms against the increase in renal vascular resistance. No less important, these observations further support the hypothesis that CGRP release and TRPV1 activation are important mediators involved in the intrarenal, capsaicin-mediated vasodilation.

Although appropriate confirmations in a human setting are lacking, in a randomized clinical trial on 21 healthy volunteers, intravesical capsaicin administration produced an increase in both the mean urinary output and the mean estimated glomerular filtration rate (eGFR; an estimation of total renal function) as well as the stimulation of natriuresis, probably by activating a vesical-renal reflex arc through the stimulation of bladder efferent activity [28]. Hence, future clinical studies are recommended for clarifying possible benefits of this molecule on renal function, particularly in the presence of chronic renal diseases.

## 4. Capsaicin Modulates Renal Nerves’ Activity

Kidneys are widely innervated organs. Adrenergic neurons supply the segments of renal vasculature and are distributed in all the segments of kidney [29]. TRPV1 channels abound in the sensory nerves of unmyelinated C-fibers or myelinated A-δ fibers innervating kidneys [30]. Hence, a direct involvement of TRPV-mediated mechanisms in the regulation of renal function appears more than plausible [4].

As previously mentioned, the activation of TRPV1 by capsaicin causes the release of neuropeptides, such as substance P and CGRP [30]. This, in turn, may activate renal afferents, causing an excitatory reno-renal reflex and sympathetic activation [31], which also involves the release of interleukin-1β from the hypothalamic paraventricular nucleus [32]. Accordingly, in an experimental model of Sprague–Dawley rats, intrarenal capsaicin infusion elicited sympatho-excitatory responses in renal nerves, which were remarkably increased during nighttime [33].

Notably, the activation of TRPV1 expressed in the unilateral renal pelvis increases the ipsilateral afferent renal nerve activity, as well as the contralateral urinary excretion of sodium and water via the reno-renal reflex [34]. In support of this hypothesis, surgical sensory nerve fibers’ denervation or degeneration impairs sodium excretion in mice kidneys treated with capsaicin, causing salt-sensitive hypertension [35].

Additionally, in dogs, the activation of the afferent C-nerve fibers by intra-arterial or intra-renal injection of capsaicin results in sympathetic excitation [36,37]. Conversely, pre-treatment with capsaicin in neonatal rats leads to a decrease or even a depletion of CGRP in the perivascular nerves of kidneys physiologically implied in the response to noxious stimuli [38], but this phenomenon is absent in adults [39]. Similarly, the administration of a capsaicin analogue, the neurotoxin resiniferatoxin, to neonatal rats reduced diuresis and natriuresis, while this did not happen in adult mice, suggesting the development of a sort of afferent fiber resistance to this treatment during adulthood [40]. This last hypothesis agrees with the fact that capsaicin-sensitive sensory neurons could be involved in the regulation of kidney function and generally supports the idea that this molecule may directly stimulate kidney nervous fibers.

Besides the above-mentioned effects, capsaicin may also regulate renal peristaltic contractions. The contractility of the renal pelvis muscles of the guineapig depends on the presence of non-adrenergic non-cholinergic innervation; the administration of capsaicin causes an initial positive inotropic response (increased contraction of the pelvis) that is not maintained after long-term administration (desensitization phenomena) [41]. The rationale behind this observation could rely on the synergic activity between TRPV1 and endothelin receptors to control the excretory function of the kidney [42]. Indeed, endothelin-1 (ET-1) is a potent vasoconstrictor and neurotransmitter found in primary afferent neurons [43], and TRPV1 colocalizes with ET-1 receptors in sensory nerve fibers innervating the renal pelvis [42], cooperating for the stimulation of diuresis and natriuresis.

## 5. Possible Beneficial Effects of Capsaicin in Kidney Diseases

The identification of novel, effective renoprotective agents for improving the treatment of renal diseases remains a largely unmet need [44]. Nowadays, promising evidence has been accumulated demonstrating different experimental benefits of capsaicin in some of the most important and complicated renal diseases, such as acute kidney injury (AKI) and diabetic kidney disease (DKD). Additionally, capsaicin may also play a protective role against renal fibrosis and pathological arterial calcifications, two hallmarks of progressive chronic kidney disease (CKD), and could partly antagonize the detrimental effects of nephrovascular and salt-sensitive hypertension (Figure 3); unfortunately, this evidence relies, again, on sparse pre-clinical models, which deserve an appropriate validation in the human setting.

### 5.1. Acute Kidney Injury

AKI is a clinical syndrome with many causes and a multifaceted pathophysiology which is defined as an acute (within hours/days) decrease in kidney function with both structural damage and function loss of the kidneys [45]. AKI complicates around 23% of the total hospitalizations worldwide, but in the intensive care unit (ICU) setting, the incidence of this condition can be as high as 78% [46]. Patient mortality due to AKI still remains dramatically high, particularly among individuals requiring dialysis support [47]. The early identification of this condition is thus crucial to initiating adequate therapeutic measures in a timely manner, thereby preventing worse patient outcomes, including the severe clinical complications, or permanent kidney damage. Unfortunately, specific therapy for AKI is lacking in the majority of cases, and preventive measures could not be as effective as expected, particularly in critically ill subjects.

There is accruing evidence indicating that capsaicin administration may prevent AKI onset in various models of kidney damage. In particular, in an in vitro model of AKI [48], capsaicin ameliorated cytotoxicity induced by lipopolysaccharides, reducing the release of specific interleukins (i.e., IL-1β and IL-18) and reactive oxygen species (ROS). Specifically, such an effect has been attributed to the activation of the TRPV1 channel and mitochondrial uncoupling protein-2 (TRPV1/UCP2) axis, triggering a protective effect against inflammation, pyroptosis, apoptosis, and mitochondrial dysfunction.

In another model of contrast-associated AKI (CA-AKI), capsaicin significantly improved tubular damage and renal dysfunction by reducing cell apoptosis, renal malondialdehyde, and superoxide, also improving mitochondrial function and structure. Notably, these effects were all mediated by an enhanced activation of the nuclear factor-erythroid 2-related factor 2 (Nrf2) [49]. In other AKI models, capsaicin was useful in preventing cisplatin- and methotrexate-induced renal damage in rats, suggesting a protective effect against toxins and lipid peroxidation as well, which represent the causative mechanisms of renal damage in this setting [50,51,52].

Sparse evidence shows that capsaicin may also ameliorate ischemic AKI, one of the most frequently observed forms of AKI in the clinical setting. The mechanism behind this beneficial effect would likely involve TRPV1 as well as TRPV4 channels, which drive an enhanced flow of calcium–potassium in endothelial cells causing vasodilation, thereby ameliorating ischemic renal injury [15].

By the same token, the activation of TRPV1, TRPV4, TRPC6, and TRPM2 on rodent models of AKI following ischemia–reperfusion promotes renoprotection through regional vasodilation [16], an observation which might endorse these surface proteins as potential therapeutic targets for ischemic AKI. Additionally, both in vitro and in vivo studies have revealed that N-octanoyl-dopamine, an agonist of TRPV1, exerts a remarkable renoprotective effect that can ameliorate AKI outcomes [53].

In mice with ischemia/reperfusion-induced kidney damage, the stimulation of TRPV1-filled primary sensory nerves by capsaicin ameliorated AKI, although the inhibition of those channels did not affect their overall outcome [54]. Conversely, other hypotheses assume that the degeneration of sensory nerves on rodent models in vivo may aggravate such a condition [52]. TRPV1 receptors are also involved in modulating inflammation and oxidative stress following ischemic kidney injury, as demonstrated in an experimental model in which rats treated with capsaicin following a salt-induced kidney ischemia and hypertension displayed a reduction in kidney damage due to the activation of TRPV1 [55].

Salt intake increases the activity of the renal sympathetic nervous system (SNS) after renal ischemia–reperfusion [56]. Mice fed with salt and treated with capsaicin show a reduction in SNS activity, an effect which can likely be attributed to the selective activation of TRPV1 channels [57].

Beyond attenuating ischemia–reperfusion-induced renal damage, preventive capsaicin administration also reduced the expression of neutrophil infiltration, renal superoxide production, and renal tumor necrosis factors (TNFs), which are all acknowledged as key players in the pathogenesis of AKI and its progression towards chronic kidney damage [58]. Evidence on the putative renoprotective effects of capsaicin in the setting of AKI is, thus, convincing (Table 1). Yet, such findings remain confined to experimental models and would need to be confirmed in the clinical setting by targeted interventional trials.

### 5.2. Diabetic Kidney Disease

Diabetes mellitus is the leading cause of end-stage kidney disease (ESKD) worldwide, accounting for more than a half of all individuals requiring chronic dialysis treatment [59]. DKD encompasses a wide spectrum of type of renal damage due to chronic diabetes, spanning from micro-vascular alterations to selective glomerular damage with severe proteinuria and rapid progression to terminal uremia. The adoption of an optimal lifestyle, blood glucose and weight control, and the use of renoprotective agents (such as RAS inhibitors, SGLT-2 inhibitors, or mineralocorticoids) remain the mainstay combined approach to preserve renal function [60]. Yet, in a large percentage of diabetic patients, such measures are ineffective in slowing down DKD’s progression towards ESKD. The search for complementary approaches for improving renoprotection in this particular setting thus remains a timely issue.

Capsaicin has already been extensively studied as a natural method to reduce pain related to diabetic neuropathy [61], but its implications regarding DKD are still an object of intense investigation.

In particular, chronic administration of capsaicin on diabetic rats increased diuresis and the urinary excretion of the epidermal growth factor (EGF) but reduced the urinary levels of N-acetyl-b-D-glycosaminidase (NAG-L), a well-known biomarker of early kidney damage in DKD [62].

Altered intracellular calcium levels and mitochondrial dysfunction are two key features of podocyte dysfunction in DKD [63]. In diabetic mice models, oral capsaicin administration attenuated renal damage in a TRPV1-dependent manner by improving the intracellular calcium balance, by reducing the transport of calcium to the mitochondria, and by decreasing mitochondria-associated membrane formation [64]. Iron overload, which is common in diabetes, may trigger or worsen DKD [65]. In an interesting experiment, chronic capsaicin administration was tested in male Wistar rats with iron overload (IOL) and diabetes mellitus + IOL [66]. Capsaicin markedly reduced kidney iron deposits by increasing the circulating levels of hepcidin, an important regulator of iron homeostasis, but had apparently no relevant effects on biomarkers of renal damage such as albuminuria, cystatin C, and beta-2-microglobulin.

Hence, more evidence is still needed in order to better understand the true implication of capsaicin in DKD. Yet, these preliminary, interesting findings can also give a concrete hope for a possible therapeutic application of this molecule in this condition.

### 5.3. Chronic Kidney Disease

CKD is the common final route of every chronic nephropathy. In fact, regardless of the different etiologies, all chronic renal diseases converge on an irreversible histological picture, represented by renal tubulointerstitial fibrosis and renal tubular atrophy, which disrupts the cellular organization and leads progressively to renal function deterioration [67]. Despite being irreversible, the velocity of CKD progression over time is variable indeed, depending on the specific nephropathy and the additional risk factors. As for DKD, lifestyle and pharmacologic efforts to counteract CKD progression may not be fully effective in a large percentage of patients, which justifies the ample ongoing research on alternative therapeutic measures.

Experimental evidence indicates that capsaicin can reduce fibrosis accumulation on different organs [68]. In two different mouse models of renal fibrosis [69], capsaicin administration reduced fibronectin and collagen depositions in kidneys with a complex action on intracellular signals pathways, involving the inhibition of the Transforming Growth Factor-β1 small mother against decapentaplegic 2/3 signaling, which is the main promoter of profibrotic mechanisms. TRPV-1 activation by capsaicin increased intracellular calcium, upregulating various protein kinases and Silent information regulator 1, which in turn enhanced the activity of endothelial nitric oxide synthase (eNOS) with following endothelium vasodilation, finally inhibiting interstitial fibrosis [70]. These findings fit well with those reported by other studies, proving that oral capsaicin may reduce renal tubular interstitial fibrosis also by targeting the TGF-β1/epithelial–mesenchymal transition (EMT) pathway [71,72,73]. Besides renal fibrosis, pathological vascular calcification also contributes to disease progression and cardiovascular complications in CKD, representing a strong predictor of mortality in these patients [74]. Chronic Hypoxic-Inducible Factor-1 alpha (HIF-1α) accumulation is known to cause osteogenic trans-differentiation, which is one of the first steps leading to diffuse arterial calcification [75,76]. In a rat model of CKD, capsaicin could inhibit the osteogenic trans-differentiation of vessels by acting either on TRPV1 activation and HIF-1α degradation through the upregulation of Sirtuin 6 [77]; such a double, synergic mechanism to prevent vascular calcification by capsaicin would absolutely deserve additional target investigations to ascertain whether this natural compound could indeed represent a valid therapeutic option for ameliorating this serious and still irreversible complication of CKD.

### 5.4. Arterial and Renovascular Hypertension

The kidney plays a determinant role in regulating blood pressure homeostasis, and deranged hormonal or vascular kidney responses have been implicated in the pathogenesis of either essential or secondary forms of arterial hypertension. On the other hand, hypertension remains one of the major risk factors for the onset and progression of kidney diseases [78]. As briefly alluded to before, CGRP is a potent vasodilator and is the principal neurotransmitter in capsaicin-sensitive sensory nerves. Besides its vasodilatory effects, this peptide is involved in the control of arterial pressure by interacting with the renin–angiotensin–aldosterone (RAS) and the sympathetic nervous system and may modulate the proliferation of the smooth muscle cells in the medium layer of arterial vessels [79]. As previously said, capsaicin is a potent inductor of CGRP release [30]. Accordingly, experimental administration of this substance can ameliorate hypertension in rat models, an effect which is partially mediated by an increased release of the insulin-like growth factor 1 [80]. Sodium excess is another fundamental player in the pathogenesis of hypertension [81]. Induction of TRPV4 channel activation causes hypotension in rats fed salt, suggesting that this receptor channel has a protective role against salt-induced hypertension [82]. According to this hypothesis, the preventive blockade of TRPV4 channels expressed in kidneys leads to a significant increase in the blood pressure values of salt-sensitive mice [83]. As previously mentioned, capsaicin also exerts a natriuretic role by activating TRPV1 channels, which promote the expression of epithelial sodium channels in the kidneys. Thus, long-term administration of capsaicin could be helpful for preventing the development of hypertension secondary to dietary salt overload [84].

As is well-acknowledged, renal denervation leads to a remarkable decrease in arterial pressure [85]; this is, at least in part, attributable to the disruption of overactive renal nerves expressing TRPV1. In fact, deprivation of those channels in rats in the presence of capsaicin caused a lack of sympathetic activity stimulation with a following reduction in blood pressure values and a significant increase in the glomerular filtration rate [86]. On the other hand, high salt intake after sensory denervation in rats increases blood pressure values, thus indicating that salt overload induces hypertension independently of sensory nervous activity [87]. Effectively, the blockade of TRPV1 causes an increase of blood pressure values in salt-resistant rats fed with a high salt diet, while it has no effect on salt-sensitive rats fed with a normal sodium diet; on the other hand, the stimulation of TPRV1 decreases blood pressure values more in salt-resistant animals fed with a high-salt diet than in others [88]. These results can prove that TRPV1 is activated during a chronic dietary salt overload, implying that this channel may play a central role in the pathogenesis of salt-sensitivity hypertension. Salt overload in salt-sensitive rats impairs the activity of TRPV1 in their kidneys, which suppresses the release of CGRP and substance P in the renal pelvis [89]. However, this does not happen in salt-resistant mice fed with high salt intake. Hence, bearing in mind that CGRP and substance P may act as vasodilators, these findings suggest that those two molecules, as well as capsaicin, which drives their release, could be helpful for preventing renovascular hypertension. The potential benefits of capsaicin in reno-vascular hypertension have also been highlighted in another experiment focusing on the vasodilatory effects of this molecule and its capacity of triggering the release of nitric oxide [90]. However, direct renal infusion of capsaicin increases the contralateral renal sympathetic nerve activity in a dose-dependent manner, which leads to a paradoxical increase in blood pressure through an excitatory renal reflex mediated by the paraventricular nucleus [91]. Additionally, the degeneration of TRPV1-filled nerves enhances salt-induced hypertension in rats after renal ischemia–reperfusion injury through the release of inflammatory mediators [92]. Taken all together, these findings indicate that TRPV1 channels could represent a promising target for the treatment of salt-sensitive renal hypertension, also suggesting a potential role for capsaicin as a natural remedy for ameliorating blood pressure control together with the use of common antihypertensive drugs (Table 2).

### 5.5. Renal Cancer

The anti-tumoral properties of capsaicin on different types of cancer cells are well-acknowledged, but definite data for recommending its daily use to synergize traditional anticancer therapy are still missing [93]. By targeting multiple signaling pathways, oncogenes, and tumor-suppressor genes, this substance may regulate the expression of different genes involved in cell survival, growth arrest, metastasis, and angiogenesis, as demonstrated in various models of cancer [94]. On top of that, capsaicin can promote changes in cell morphology and migration, probably by impacting cell-to-cell interactions, cell migration, and cell morphology; such effects would be likely driven by its interaction with the vanilloid receptors and the following regulation of calcium flow [95]. In a milestone experiment, capsaicin demonstrated a significant capacity of inhibiting migration and the invasion of renal cancer cells both in vitro and in vivo, as well as promoting cellular autophagy by activating the AMPK/mTOR pathway [96]. Such observations gave concrete support to the potential therapeutical application of this substance as an inhibitor of renal cancer invasion and peripheral metastasis. In addition, capsaicin promotes the inhibition of the PD-L1/PD-1 checkpoint, limiting the proliferation of human bladder and renal cancer cells [97]. In another model, capsaicin displayed an undisputable anticancer activity on human renal carcinoma by inducing apoptosis through the p38 and JNKs/MAPKs pathways, which are implied in the control of cell cycle progression [98]. Despite this preliminary evidence, however, the true anticancer effect of capsaicin on human renal neoplasias deserves an appropriate confirmation by focused clinical studies.

## 6. Conclusions

Capsaicin has been extensively studied for years for its unique physiochemical properties and the biological effects exerted on different tissues and organ systems. Nowadays, interesting but very sparse evidence is accumulating, pointing at this substance as a biologically active factor in kidneys as well, particularly with respect to the hemodynamic, nervous, and functional effects described in renal structures after experimental administration. Additionally, capsaicin may serve as a protective factor against different types of kidney injuries and diseases, including hypertension, which endorses this molecule as a potential renoprotective agent, besides the traditional therapies, where available. Yet, to date, clinical evidence on such benefits is lacking, particularly in the scenario of human kidney diseases, in which the only clinically approved indication of capsaicin use is limited, nowadays, to the topical treatment of uremic pruritus [99], a largely prevalent complication of uremia which represents a source of stress and of a reduced quality of life [100].

Unfortunately, notwithstanding the above-mentioned benefits of capsaicin in systemic disorders, the practical usability of this substance as a therapeutic agent remains hampered by various limitations. As a matter of example, following chronic exposure to capsaicin, a “desensitizing” effect attributable to a tolerance phenomenon may manifest, particularly in habitual consumers of capsaicin-enriched foods [101]. Such an event may reduce the magnitude of benefits over time, thereby limiting the efficacy of long-term treatments in chronic diseases. Furthermore, due to its chemical structure, capsaicin has low hydrophilicity, scarce oral bioavailability, and is poorly absorbed because of an important first-pass metabolism in the liver and its poor aqueous solubility. The use of capsaicin-filled nanocapsules has been proposed as a suitable solution to maximize gastrointestinal absorption; unfortunately, to this extent, larger doses of capsaicin are needed to equal the biological effects of the pure compound [102], which may increase the risk of adverse events. These could include irritation of the mucous gastrointestinal layer which may cause nausea, vomiting, and burning diarrhea or even the appearance of peptic ulcers. Furthermore, in allergic individuals, high-dose exposure to capsaicin may also induce severe bronchoconstriction due to the greater activity of TRPV1 receptors at the pulmonary level [103]. Hence, although capsaicin is commonly considered safe, relevant side effects in predisposed individuals should always be taken into consideration.

Still, this natural substance remains a very promising natural approach for preventing or treating various kidney disorders, as indicated by sparse although concordant experimental findings. Future clinical interventional studies are thus eagerly advocated for to demonstrate the potential practical usefulness of capsaicin for improving renal health and also slowing diseases’ progression in daily clinical practice.

## Figures and Tables

**Figure 1 ijms-25-00791-f001:**
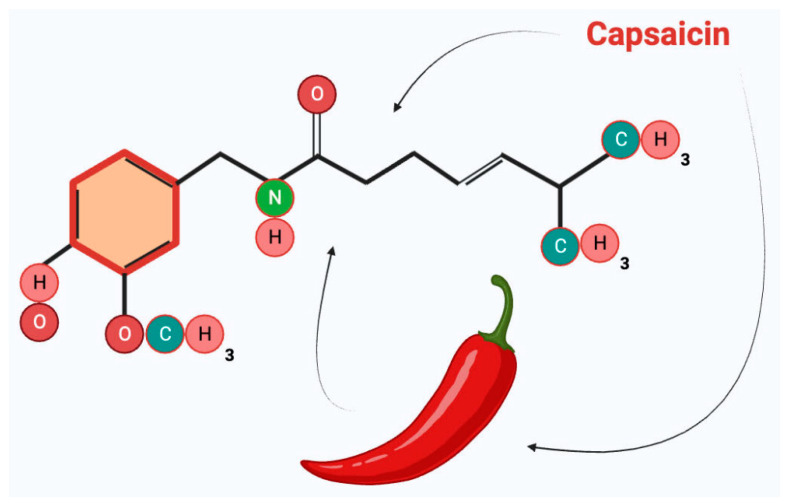
Chemical structure of capsaicin.

**Figure 2 ijms-25-00791-f002:**
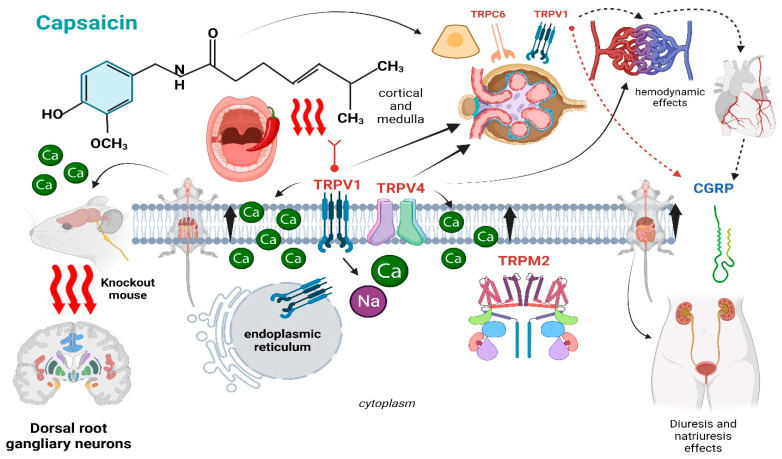
Summary of the main physiological effects of capsaicin on the kidney.

**Figure 3 ijms-25-00791-f003:**
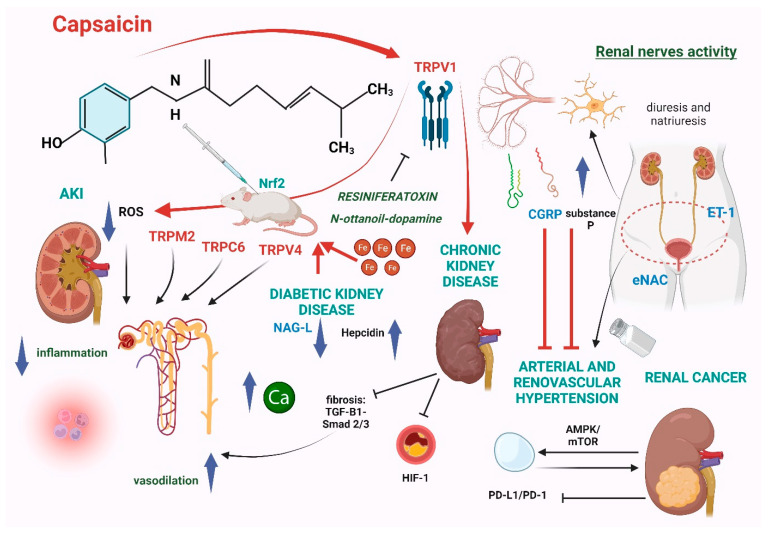
Putative mechanisms underlying the benefits of capsaicin administration on kidney diseases and hypertension.

**Table 1 ijms-25-00791-t001:** Main experimental studies testing the effects of capsaicin in different models of AKI.

Authors	Models	Results
Han et al. [48]	HK-2 cells treated with ATP and LPS	Capsaicin preincubation ameliorated LPS-induced cytotoxicity through TRPV1/UCP2 axis activation by reducing IL-1β, IL-18, and ROS release.
Ran et al. [49]	Dehydrated C57BL/6J mice treated with the contrast medium iodixanol	Preventive capsaicin administration reduced contrast-induced AKI through Nrf2 activation by decreasing superoxide, renal malondialdehyde, and apoptotic tubular cells and improving mitochondrial function.
Shimeda et al. [51]	Male Sprague–Dawley rats treated with cisplatin	Dietary capsaicin reduced cisplatin-induced renal damage by reducing lipid peroxidation.
Aldossary et al. [50]	AKI following methotrexate intoxication in rats	Capsaicin administration reduced methotrexate-induced renal damage by anti-inflammatory and antioxidant effects.
Tsagogiorgas et al. [53]	Inbred male Lewis rats treated with NOD	Treatment with the synthetic analogue of capsaicin NOD had renoprotective effects against ischemia-induced AKI through TRPV1 activation by inhibiting TNF-α mediated inflammation and through production of the vasodilator peptides CGRP and SP.
Yu et al. [55]	Male Wistar rats fed with high-salt diet	Capsaicin injection reduced renal inflammation driven by high-salt diet, oxidative stress, and fibrosis through activation of TRPV1.
Yu et al. [57]	Rats fed with high-salt diet after ischemia–reperfusion damage	Capsaicin inhibited renal sympathetic nerve activity by activating TRPV1 receptors, which prevented the appearance of salt sensitivity following renal ischemia–reperfusion damage.
Ueda et al. [58]	Uninephrectomized male Sprague–Dawley rats developing AKI following renal artery and vein occlusion	Treatment with capsaicin or its analogue resiniferatoxin reduced ischemia–reperfusion renal damage by reducing neutrophil infiltration, superoxide production, and TNF-α production and by increasing IL-10 production.

Legend: AKI: acute kidney injury; ATP: adenosine triphosphate; CGRP: calcitonin gene-related peptide; HK-2: human kidney 2; IL-1β: interleukin-1 beta; IL-10: interleukin-10; IL-18: interleukin-18; LPS: lipopolysaccharide; NOD: N-octanoyl-dopamine; Nrf2: nuclear factor erythroid 2-related factor 2; ROS: reactive oxygen species; SP: substance-P; TRPV1: transient receptor potential vanilloid type 1; UCP2: uncoupling protein 2; TNF-α: tumoral necrosis factor alpha.

**Table 2 ijms-25-00791-t002:** Main studies testing the benefits of capsaicin administration in different models of reno-vascular hypertension.

Authors	Model	Results
Harada et al. [80]	Spontaneously hypertensive rats and Wistar Kyoto rats	Capsaicin administration increased CGRP and IGF-1 plasma levels in SHR as compared to those reported in WKR.
Gao et al. [82]	Male Wistar rats fed with normal sodium diet and high sodium diet	HS diet induced TRPV4 expression in mesenteric arteries and sensory nerves with following increase in CGRP and IGF-1 levels. HS diet induced a marked increase of blood pressure when TRPV4 channel was blocked.
Li et al. [84]	C57BL/6 wild-type mice and TRPV1-/- mice	Dietary capsaicin induced natriuretic effect by inhibiting WNK1/SGK1/aENaC pathway with consequent reduction of aENaC expression at the renal level. Dietary capsaicin reduced HS diet-induced hypertension through TRPV1 activation.
Stocker et al. [86]	2-kidney-1-clip (2K1C) wild-type rats and 2K1C TRPV1-/- rats	TRPV1 channels deprivation in presence of capsaicin caused reduction in blood pressure and increase in the glomerular filtration rate due to the lack of sympathetic activity.
Ye et al. [91]	Spontaneously hypertensive rats and Wistar Kyoto rats	Renal infusion of capsaicin increased contralateral renal sympathetic nerve activation, causing an increase in blood pressure through a renal nerve reflex mediated by the paraventricular nucleus.
Segawa et al. [90]	2K1C rats and sham-operated rats	Dietary capsaicin reduced nephrovascular hypertension by promoting phosphorylation of Akt and eNOS, thus enhancing NO release.

Legend: Akt: Ak strain transforming, also known as PKB protein chinase B; CGRP: calcitonin gene-related peptide; eNOS: endothelial nitric oxide synthase; IGF-1: insulin-like growth factor 1; HS: high sodium; SHR: spontaneously hypertensive rats; 2K1C: 2-kidney-1-clip; NO: nitric oxide; TRPV1-/-: transient receptor potential vanilloid type 1 knock-out rats; WKR: Wistar Kyoto rats.

## Data Availability

No new data were created or analyzed in this study. Data sharing is not applicable to this article.

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
