# Peer review of "Spice Up Your Kidney: A Review on the Effects of Capsaicin in Renal Physiology and Disease"

_ijms, 2024, doi:10.3390/ijms25020791_

Round 1
Reviewer 1 Report
Comments and Suggestions for Authors
Authors should revise the review accordingly

Minor typo-errors.
Author Response
REVIEWER 1
This review overviews the possible role of capsaicin in kidney physiology and its associated diseases. The review is well structured and figures provide a suitable schematic representation.
-We thank the reviewer for this comment and for the time spent in reviewing our paper.
Major Comments
- Please re-write the abstract. Entire abstract presents only the background information. Abstract should also highlight the need of the review, novelty of the review and importantly what the review is offering to the readers.
-We thank the reviewer for pointing out this aspect. Following the suggestion, we have now completely reformulated the abstract accordingly.
- Present high-quality figures. Figures provided are blurry and in some instances text is not clearly readable.
-Figures have been provided embedded in the text as requested by the journal submission format and also separately in high-quality, high-res format during the upload process. We are not sure what could be the issue here. From our side, we can visualize figures in the text in full resolution and in a fully readable way. Perhaps something went wrong with the final pdf elaboration during the upload process. So, we are kindly asking the E.O. to provide a solution to solve this problem to let the reviewer appreciate the figures in full-resolution.
- Include the limitations of capsaicin in real time applications.
-We thank the reviewer for this suggestion. Limitations hampering the possible use of capsaicin in daily practice have now been elaborated in the conclusion section, focusing on possible side effects and difficulties in obtaining formulations for daily use.
- Authors should also include future prospects (if possible) for capsaicin in treating kidney associated diseases or injuries.
-Future prospects of capsaicin administration for treating kidney disorders have been summarized and emphasized in the conclusions section. Unfortunately, given the lack of clinical evidence, all applications are only speculative. We have thus emphasized the need of human trials to validate the promising findings reported by experimental models.
- Include the in-depth systematic analysis and interpretation of research articles. Research articles with animal model studies should be discussed. (https://doi.org/10.1155/2022/1763922; https://doi.org/10.1016/j.cbi.2022.110043; https://doi.org/10.3390/molecules27227764; https://doi.org/10.1016/j.kint.2023.04.023; https://doi.org/10.1152/ajpheart.00665.2021)
-We thank the reviewer for this input. The above indicated studies have now been described and commented in a more extensive and detailed way. Furthermore, we would like to point out that additional tables have now been added to summarize results from the most important experiments mentioned in the text, included the ones above mentioned.
- Try to tabulate the highlights of similar review articles and difference of this review within the same table for clarity.
-We thank the reviewer for this suggestion. However, to the best of our knowledge there are no similar reviews published on this specific issue (“capsaicin and the kidney”) , which makes impossible to compare highlights and differences with the present review. On the other hand, there is a myriad of reviews summarizing the effects of capsaicin in other settings (e.g. on rheumatic diseases, tumors, neurological pain…) which, however, falls largely beyond the topic of our review. In this narrative review, by purpose, we wanted to focus only on the effects and potential benefits of capsaicin at the kidney level.
- Only 26 out of 93 are recent publications (on or after 2020). Please include more recent references.
-Unfortunately, the majority of original articles fitting with the topic “capsaicin and kidney” are not very recent. The whole literature has been approached through a systematic search on the Medline database and we were not able to find additional, more recent evidence beyond that provided. On the other hand, we have now added ten new references to general articles (focusing on Capsaicin od kidney diseases) which have been chosen among the most recent ones.
- In section 2, molecular structure of capsaicin was provided (Figure 1). Please provide an elaborate discussion on how the structure implicitly plays a significant role in its interaction with diseased cells and their role in treatment.
-We thank the reviewer for this comment. Details on the structural interactions between capsaicin and the TRPV channels, which modulate the majority of its activities in either physiology or disease, have been provided. With respect to the single disease models, we have further expanded the details of the cellular mechanisms underlying the benefits of capsaicin, whereas provided by the individual studies.
Minor Comments
- References should be journal format.
-All references have now been reformatted using the established journal style with a reference manager software (EndNote)
- Use abbreviations accordingly. Abbreviations should be used at first mention. For example, CKD at line 195 and also abbreviated in line 299. If it is difficult to follow through the use of abbreviations, authors can present them as a list at the beginning only.
-The paper has now been re-checked and all abbreviations have been explained only at first mention
Remark
The review is well written but overall provides only basic information. Lack of tables is major drawback. Review also lacks proper systematic and constructive discussion on recent research articles. The review needs to properly revise to include more detail oriented discussion on provided sections.
-As specified in the previous responses, we have tried to address as many suggestions as possible in order to improve the readability of the paper, going much more into details of some experimental models and adding other sections to better clarifying some concepts. Additionally, to overcome the absence of tables, we have now added two tables summarizing the main evidence from experimental studies of capsaicin in the context of AKI (table 1) and nephrovascular hypertension (table 2). The abstract, conclusions and the descriptive part of molecular mechanisms of capsaicin have all been improved, following the reviewer’s suggestion. Moreover, additional references have been inserted. We sincerely hope that all the efforts we have made to ameliorate the quality of our paper will be appreciated by the reviewer.
Reviewer 2 Report
Comments and Suggestions for Authors
Your manuscript lacks the novelty and originality that are necessary for publication in IJMS. I can't see any major new aspect in your paper that hasn't been addressed already. Furthermore, the manuscript frequently reads more like a qualitative report than a critical review, which would summarize, analyze, and assess information from the literature that has already been published. On the other hand, the scientific quality of this review is very low. Due to lack of novelty, I do not recommend publication of this manuscript in a high prestige journal like IJMS
Comments on the Quality of English LanguageLanguage correction is needed.
Author Response
REVIEWER 2
Your manuscript lacks the novelty and originality that are necessary for publication in IJMS. I can't see any major new aspect in your paper that hasn't been addressed already. Furthermore, the manuscript frequently reads more like a qualitative report than a critical review, which would summarize, analyze, and assess information from the literature that has already been published. On the other hand, the scientific quality of this review is very low. Due to lack of novelty, I do not recommend publication of this manuscript in a high prestige journal like IJMS
-We thank the reviewer for the time spent in reading our paper and we sincerely regret that the overall aim of our review has not been properly understood. Currently, there are myriad of reviews dealing with the effects, benefits, harms, potential applications, future directions of capsaicin in various settings, spanning from oncology to rheumatic diseases, neurological pain and even cardiovascular disease. To the best of our knowledge, however, no review has been published so far summarizing the renal effects of this molecule which, indeed, are numerous and potentially useful also in the context of treating some renal diseases. Hence, we feel that this is a strong element of novelty which distinguish our review from the several others regarding capsaicin. As nephrologists, we were really interested to get more insights on this aspect, and we felt that it would have been interesting to share this summarized knowlegde on the pleiotropic effect of capsaicin at the kidney level (perhaps less intriguing that those exerted on other organs for a general audience but still remarkably interesting) with the readership of the journal to give a snapshot of less known effects of this widely used substance. Again, we sincerely regret that this message has not reached this reviewer. Following the other reviewers’ suggestions, we now have improved the overall scientific quality of our paper, including a more critical part, additional references, tables and reformulated conclusions. We hope that the reviewer may give another chance to our work and, perhaps, appreciate the efforts made to improve it.
Reviewer 3 Report
Comments and Suggestions for Authors
In this review, the authors explore capsaicin, the spicy compound in red peppers, which is under scrutiny for its natural remedy potential in cancer and cardiovascular diseases. They describe research that hints at kidney benefits, impacting function, vasorelaxation, and hypertension. While promising for preventing kidney issues and cancer growth, human trials are essential for conclusive evidence.
The article has an interesting topic and an attractive title. The figures are very helpful and engaging.
The introduction was well-written, covered the main background points, and led up to the aim of the review.
However, small issues were detected.
First of all, the first part of the article is very well written, but somewhere on page 8, the quality drops noticeably. Unlike the rest of the text, from this point on the paragraphs become too big, which makes them difficult to follow. Page 9 is a wall of text ...
As for the references, they seem to be omitted in a number of places:
The first part of the introduction has only one reference. A review article. It would be appropriate to cite the original articles used in that review. There are similar omissions in the rest of the article.
Sentence 38-39 is missing references.
Line 45-47 is missing reference.
Line 86-88 is missing references....There are similar omissions in the rest of the article.
There is a question mark in brackets in line 114.
There is a space missing in line 91.
Some paragraphs do not have a single reference. Please add references appropriately.
All in all, I do not have any major concerns and will gladly accept the article after these oversights have been addressed.
Author Response
REVIEWER 3
In this review, the authors explore capsaicin, the spicy compound in red peppers, which is under scrutiny for its natural remedy potential in cancer and cardiovascular diseases. They describe research that hints at kidney benefits, impacting function, vasorelaxation, and hypertension. While promising for preventing kidney issues and cancer growth, human trials are essential for conclusive evidence. The article has an interesting topic and an attractive title. The figures are very helpful and engaging. The introduction was well-written, covered the main background points, and led up to the aim of the review.
-We sincerely thank the reviewer for the positive comments, for the time spent in reading and appraising our review and we are happy that our efforts have been appreciated.
However, small issues were detected.
First of all, the first part of the article is very well written, but somewhere on page 8, the quality drops noticeably. Unlike the rest of the text, from this point on the paragraphs become too big, which makes them difficult to follow. Page 9 is a wall of text ...
-We thank the reviewer for underlining this aspect. We agree that the readability might be hampered in the latter sections as these parts report a series of elaborated experimental studies and molecular mechanisms. In order to improve clearness of this part of the manuscript, we have now added a summary table depicting the main results from the studies mentioned. We hope that this could improve the readability of this section of the review.
As for the references, they seem to be omitted in a number of places:
The first part of the introduction has only one reference. A review article. It would be appropriate to cite the original articles used in that review. There are similar omissions in the rest of the article.
Sentence 38-39 is missing references.
Line 45-47 is missing reference.
Line 86-88 is missing references.There are similar omissions in the rest of the article.
There is a question mark in brackets in line 114.
There is a space missing in line 91.
Some paragraphs do not have a single reference. Please add references appropriately.
-We thank the reviewer for pointing out this issue. The whole manuscript has been carefully checked and some sentences lacking appropriate references have indeed been identified. In some cases, those sentences were referred to experimental studies mentioned just a couple of lines above. In the remaining, we have now provided additional pertinent references. In total, we have added 10 more citations with respect to the previous version. Additionally, minor typo errors (included the ones mentioned) have been identified and corrected.
All in all, I do not have any major concerns and will gladly accept the article after these oversights have been addressed.
Round 2
Reviewer 2 Report
Comments and Suggestions for Authors
Now the manuscript is may be accepted for publication
Comments on the Quality of English LanguageManuscript is now improved